# Protocol for individual participant data meta-analysis of randomised controlled trials of patients with psychosis to investigate treatment effect modifiers for CBT versus treatment as usual or other psychosocial interventions

Maria Sudell [1], Catrin Tudur-Smith [1], Xiaomeng Liao,[1] Eleanor Longden,[2,3] Graham Dunn,[4] Tim Kendall,[5] Richard Emsley [6], Anthony Morrison [2,7], Filippo Varese [3,7]

For numbered affiliations see end of article.

**Correspondence to**
Dr Maria Sudell;
mesudell@liverpool.ac.uk

## ABSTRACT

**Introduction** Aggregate data meta-analyses have shown heterogeneous treatment effects for cognitive–behavioural therapy (CBT) for patients with schizophrenia spectrum diagnoses. This heterogeneity could stem from specific intervention or patient characteristics that could influence the clinical effectiveness of CBT, termed treatment effect modifiers. This individual participant data meta-analysis will investigate a range of potential treatment effect modifiers of the efficacy of CBT.

**Methods and analysis** We will perform a systematic review and meta-analysis of studies investigating CBT versus treatment as usual, or CBT versus other psychosocial interventions, for patients with schizophrenia spectrum diagnoses. The Cochrane Central Register of Controlled Trials (CENTRAL), PubMed, EMBASE and the online clinical trials registers of the US government, European Union, WHO and Current Controlled Trials will be searched. Two researchers will screen titles and abstracts identified by the search. Individual participant data will be requested for any eligible study, for the primary outcome (overall psychotic symptoms), secondary outcomes and treatment effect modifiers. Data will be checked and recoded according to an established statistical analysis plan. One-stage and two-stage random effects meta-analyses investigating potential treatment effect modifiers will be conducted. A list of potential treatment effect modifiers for CBT will be produced, motivating future research into particular modifiers.

**Ethics and dissemination** This study does not require ethical approval as it is based on data from existing studies, although best ethical practice for secondary analysis of clinical data will be followed. The findings will be submitted for publication in peer-reviewed journals, and promoted to relevant stakeholders.

**PROSPERO registration number** CRD42017060068.

## INTRODUCTION

Cognitive–behavioural therapy (CBT) is a recommended intervention for the treatment and management of psychosis and

### Strengths and limitations of this study

► This will be the first published individual participant data meta-analysis to investigate treatment effect modifiers for cognitive–behavioural therapy (CBT) for patients with schizophrenia spectrum diagnoses.
► The review will consider the efficacy of CBT across multiple outcomes of interest in addition to psychotic symptoms severity.
► The search will be conducted without geographical, language or time restrictions.
► A potential limitation of this study will be the degree of heterogeneity between studies in recorded measures and scales employed.

schizophrenia.[1] Aggregate data meta-analyses (AD-MA) suggest that CBT for psychosis has modest but considerably heterogeneous treatment effects (eg, ref 2 3). This inconsistency partly stems from intertrial variation in several key methodological characteristics of the existing randomised controlled trials (RCT; eg, blinding/masking of outcome assessments[2 3]) but could also reflect the impact of unaccounted clinical heterogeneity, that is, specific intervention and patient characteristics that can potentially influence the clinical effectiveness of CBT. For instance, previous trials differed widely in terms of intervention characteristics (eg, number of treatment sessions, treatment duration, use of manualised interventions), patients' baseline severity of psychotic and other comorbid symptoms, their demographic characteristics (eg, age, gender and ethnic origin) and illness duration. The identification of moderators of treatment response and/or

subgroups of patients who may particularly benefit from CBT would allow optimisation of treatment delivery, with significant implications in terms of improved clinical effectiveness, cost savings and maximisation of patients' informed choice of treatment.

The impact of these potential treatment effect modifiers, however, remains unaccounted for (or at best poorly estimated) in AD-MA due to their reliance of the reporting quality of primary studies and the limited statistical power of 'standard' meta-analytical methods for testing treatment effect moderators.[4] Additional primary research would be costly and impractical given the large sample size required. The only approach suited for this type of research is individual participant data meta-analysis (IPD-MA), a research synthesis method which summarises the evidence on a particular clinical question by considering individual participant-level rather than aggregate-level data from multiple related studies. IPD-MA allows (1) greater ability to examine the impact of multiple individual-level and study-level factors (and their combination) on the treatment effects considered, (2) standardisation of statistical methods used across studies, and (3) the potential of reduced risk of bias, for example, due to selective reporting of outcomes compared with conventional AD-MA.[5–8] This article is the protocol for an IPD-MA to assess the impact of treatment effect modifiers of CBT.

## METHODS
### Objectives of IPD-MA
#### Primary outcome
The primary objective of the IPD-MA of RCTs is to identify treatment effect modifiers for CBT or CBT+ versus treatment as usual (TAU) or other psychosocial interventions on overall psychotic symptoms severity as measured by Positive and Negative Syndrome Scale (PANSS) scores[9 10] in patients with schizophrenia spectrum diagnoses.

#### Secondary outcomes
In this IPD-MA of RCTs, treatment effect modifiers for CBT or CBT+ versus TAU or other psychosocial interventions will also be examined for patients with schizophrenia spectrum diagnoses for the secondary outcomes listed in box 1.

For some secondary outcomes, multiple suboutcomes will be examined, such as change in severity of affective symptoms, which is examined both for anxiety and depression. Adverse event information will be sought specifically for the four suboutcomes mentioned, but additional adverse event information will also be tabulated. Secondary outcomes were selected based on (1) outcomes often targeted in CBT for psychosis, (2) measures identified as valuable in our patient and public involvement (PPI) consultations.

---

**Box 1    Secondary outcomes**

Minimum clinical important differences (MCIDs) in Positive and Negative Syndrome Scale (PANSS) scores[9 10 13 27]
► Reduction of ≥11 points.
► Reduction of ≥15 points.
Clinically significant deterioration (CSD) in PANSS scores[9 10 13 27]
► Increase of ≥11 points.
► Increase of ≥15 points.
Change in specific symptom clusters
► Positive psychopathology.
► Negative psychopathology.
► General psychopathology.
Change in specific symptoms often targeted in cognitive–behavioural therapy (CBT) for psychosis
► Hallucinations severity.
► Delusions severity.
► Hallucination-associated subjective distress.
► Delusion-associated subjective distress.
► Paranoia severity.
Change in severity of affective symptoms
► Anxiety.
► Depression.
Subjectively defined recovery
Quality of life
Social and occupational functioning
Early treatment discontinuation
Adverse effects
► Deaths.
► Attempts at suicide.
► Suicide ideation.
► Serious violent incidents.
Hospital readmissions

---

### Treatment comparisons
Consistent with the analytical approaches in recent AD-MA[11 12] our IPD-MA will distinguish between 'pure' CBT interventions as defined by the National Institute for Health and Care Excellence (NICE)[1] and 'CBT+' interventions, where CBT+ is defined as a CBT treatment package incorporating significant elements of other distinct psychosocial intervention approaches (eg, mindfulness, motivational interviewing, family intervention) alongside core CBT elements. CBT and CBT+ will be compared with TAU or other psychosocial interventions in individuals with a diagnosis of schizophrenia spectrum disorders. TAU is defined as the level of care service users would routinely receive had they not been involved in the trial, other psychosocial interventions is defined as TAU supplemented by additional psychological or social interventions, for example, family therapy. Although all trials of CBT against these comparators will be eligible, these will be synthesised and contrasted in separate analyses for: (1) CBT versus TAU, (2) CBT versus other psychosocial interventions, (3) CBT+ versus TAU, and (4) CBT+ versus other psychosocial interventions.

### Treatment effect modifiers
The selection of treatment effect modifiers was informed by (1) knowledge of variables examined in previous and

**Table 1** Treatment effect modifiers examined in this IPD-MA

| | Treatment effect modifier |
|---|---|
| Participant's demographic characteristics | Age at entry to trial |
| | Gender |
| | Ethnicity |
| Participant's clinical characteristics | Effect of specific diagnostic subgroups |
| | Phase of illness (first-episode psychosis/multiple-episode psychosis) |
| | Illness duration |
| | Duration of untreated psychosis |
| | Initial severity of psychotic symptoms (measured by baseline PANSS scores) |
| | Initial severity of comorbid affective symptoms (measured by baseline anxiety scores) |
| | Initial severity of comorbid affective symptoms (measured by baseline depression scores) |
| | Dosage equivalence of baseline antipsychotic medication(s)[27] |
| | Number of antipsychotic medications received at baseline |
| Specific intervention characteristics | Time period over which treatment was delivered* |
| | Number of therapy sessions offered in the study* |
| | Number of therapy sessions attended by the individual |
| | Minimum study required level of therapist's training and competence* |
| | Measures of therapeutic alliance |
| | Use of manualised interventions* |
| | Use of formulation-based interventions* |
| | Indicator for whether the intervention was designed to target the outcome under scrutiny* |
| | Individual versus group interventions* |
| | Length of study follow-up |

*Treatment effect modifiers which are study-level variables, the remaining are individual-level variables.
IPD-MA, individual participant data meta-analysis; PANSS, Positive and Negative Syndrome Scale.

ongoing RCTs by members of our team and the collaborators in the CBTp: IMPART (Individual Modifiers of PAtient Response to Treatment) Consortium; (2) the findings of studies which examined predictors of outcomes in previous RCTs (eg, ref 13–16), and (3) consultation meetings with service users with psychosis and clinical psychologists and CBT therapists working with clients with psychosis in secondary care settings in the UK.

The treatment effect modifiers shown in table 1 will be investigated for both the primary and secondary outcomes. Treatment effect modifiers have been separated into three main groups: (1) participant demographic characteristics, (2) participant's clinical characteristics, and (3) specific intervention characteristics.

### Protocol
This evidence synthesis will follow state-of-the-art guidelines for IPD-MA, and our outputs will comply as a minimum with the Preferred Reporting Items for Systematic Reviews and Meta-Analyses statement for the reporting of IPD-MA.[8] The search strategy of our IPD-MA builds on the protocol and database searches carried out as part of a recent AD-MA carried out by members of our team.[11 12] Our study selection criteria are consistent with those employed in this recent AD-MA. Similarly, our literature searches will update those carried out as part of this review to identify any RCTs that have become available since the date of search.

### Search strategy
We will update the searches already conducted as part of a recent AD-MA[11 12] to identify any trials that might have been become available for research synthesis since the date of last search. Database searches will be conducted on the Cochrane Central Register of Controlled Trials (CENTRAL), PubMed, EMBASE and the online clinical trials registers of the US government, European Union, WHO and Current Controlled Trials.

In line with the protocol of the AD-MA carried out by members of our team,[11 12] titles, abstracts and keywords will be searched in the publication databases using the adaptations of the following generic strategy: (schizo$ [exp. schizophrenia+psychosis+schizoaffective]) AND (trial [exp. RCT+controlled trial+clinical trial]) AND (cbt [exp. cognitive therapy+behaviour therapy+psychotherapy]).

### Selection of studies
The project's principal investigator (PI) and another member of the research team will screen titles and abstracts for relevance, and subsequently assess eligibility by examining the full-text reports against the above-mentioned criteria. When required, additional information to ascertain eligibility will be requested from the RCT authors and through the examination of treatment manual when available (eg, to ascertain that interventions complied with NICE operational definitions of CBT). Discrepancies in selection decisions will be discussed, and arbitration by other members of the research team sought to achieve consensus. We will include studies based on the following eligibility criteria.

### Participants
Trials where >50% of participants have diagnoses in the schizophrenia spectrum (schizophrenia, schizoaffective disorder or early psychosis) will be eligible. Trials where >50% of participants have an established diagnosis of bipolar disorder, intellectual disability, psychosis

secondary to a general medical condition or organic pathology, or a primary diagnosis of substance-induced psychosis will be excluded. No restriction will be placed on participants' age, ethnicity, illness severity and illness duration.

## Treatment comparisons

Included studies must compare treatments that fit into one or more of the four stated treatment categories (CBT, CBT+, TAU, other psychosocial interventions). Study interventions will be classified into one of these four categories by clinicians involved with this project.

## Outcomes

The study must provide data for one or more of the stated primary or secondary outcomes. Data can be recorded on any comparable scale at any time point. Data will also be sought for the treatment effect modifiers listed in table 1.

## Study designs

Parallel single-blind or open controlled trials with at least two arms using random allocation to treatment will be considered for inclusion. Single-arm or cross-over studies will not be eligible for inclusion in the review. Studies employing other research designs (case series, cohort analyses) will not be eligible.

Trials will be eligible if they evaluated CBT or CBT+ interventions versus TAU or other psychosocial treatments, eligible treatments as defined in the Treatment comparisons section.

## Data collection and processes

IPD will be collected from PIs of past and ongoing RCTs of CBT for psychosis in the UK and internationally. Our ability to successfully collect relevant IPD is facilitated by several factors. Our research team includes researchers who have conducted some of the largest RCTs in this area. Furthermore, we have established a network of collaborators to support the retrieval of relevant IPD: the CBTp: IMPART Consortium. We will continue to expand the CBTp: IMPART Consortium over the lifetime of the project by sending invitation emails to all researchers who have published and/or are currently conducting RCTs relevant to this work. Participating researchers will be sent specific data request forms outlining variables pertinent to the present IPD-MA. They will be asked to fully anonymise the requested data set and share them with our research team using a safe data transfer system provided by the information technology services at the University of Liverpool. We will remain in regular contact with all Consortium members throughout the lifetime of the project to clarify queries about their IPD and its integrity.

In the case of no response to data requests (defined as a minimum of four contact attempts with no response), details of the study would be stated as 'non-acquired data'. Details concerning the number and proportion of studies and individuals for which IPD has been obtained will be stated in reports of analyses. Additionally, we have planned sensitivity analyses to assess the impact of non-acquired data on results of the project, which will combine obtained IPD, with AD from studies not supplying IPD.

A statistical analysis plan detailing the data cleaning and coding, and the analyses to be conducted has been produced and will be available on request. Data received will be systematically recoded to ensure common scales or measurements across studies. We will liaise with PIs and statisticians of the primary studies to resolve any data issues and prepare the data set for IPD-MA. In addition to primary and secondary outcome data, IPD and relevant supporting material (eg, trial codebook, therapy manuals, statistical analysis plans) will be requested to code the stated treatment groups, treatment effect modifiers, and primary and secondary outcomes.

The primary outcome will be analysed on the PANSS, with comparable information recorded on other scales converted onto the PANSS where possible (eg, Leucht *et al*[17] provide supplemental tables to convert between PANSS and Brief Psychiatric Rating Scale). Secondary outcomes will be transformed onto the most commonly reported scale, where established conversion tables exist. If transformation to a common scale is impossible, or there is no most common scale, standardised values (calculated by dividing by the between-patient variation) will be employed.[18] Transformation onto subscales will not be attempted. For example, secondary outcome change in specific symptom clusters: positive is defined as the sum of the positive subscales of the PANSS score. If a study records a comparable measure, no attempt will be made to transform this measure onto the positive subscale of PANSS; however, the data will contribute to a standardised score analysis, for example, as part of a sensitivity analysis.

## Statistical analysis
### Data analysis

All randomised patients will be included, and an intention-to-treat principle will be followed throughout. To examine IPD integrity and concordance with original trial analyses, all trials will be reanalysed individually and the original authors asked to confirm the individual study results and resolve any discrepancies. Throughout, a frequentist approach to analyses will be taken. All analyses will be conducted using R.[19] If the planned quantitative analyses cannot be undertaken, the data will be described qualitatively. We will examine the pooled treatment effect for each outcome by performing a series of one-step IPD-MA (where IPD from individual studies are analysed simultaneously while accounting for study-level clustering) and two-step IPD-MA (where estimates of the treatment effect are initially computed from the IPD of each study, and then aggregated using conventional inverse variance meta-analytical approach).[5] Throughout, due to anticipated heterogeneity between studies, a random effects approach will be employed (using a DerSimonian and Laird approach for two-stage analyses,[20] and including

a study-level random treatment effect, and fixed study membership effect in one-stage approaches).

Both one-stage and two-stage analyses will be conducted to allow a full investigation of the data.[21] The two-stage analyses will be conducted initially to help identify areas of higher between-study heterogeneity through examination of, for example, forest plots of analyses. One-stage analyses will then be conducted to allow multiple interactions between treatment modifiers to be examined. One-stage and two-stage analysis results will be compared to confirm that areas of heterogeneity are identified similarly between each approach.

Due to considerable variation in the follow-up periods considered in the original trials, separate analysis of trials with highly comparable or identical points of assessment (eg, 3 months, 6 months, etc) would be unfeasible. In order to maximise the data contributing to the analysis, outcomes measured at multiple time points will be modelled longitudinally. As dropout may be an issue during study periods, we will employ joint modelling methods (eg, ref [22] [23]) to account for study dropout. Outcomes measured at a single time point (eg, at the end of the treatment period) will be analysed using generalised linear models (GLM).

In two-stage analyses, treatment modifier interactions will be estimated within each study, and the results pooled. In one-stage analyses, treatment moderator interactions separating out within-study and between-study effects will be examined, while accounting for clustering of participants within studies.[5 6 24] Any treatment effect modifier found to be significant at a level of 0.05 will be retained in a list of potential treatment effect modifiers. Treatment effect modifiers with a significant effect for each outcome will be identified through examination of 95% CIs. CIs for both joint and GLM analyses will be calculated through bootstrapping using 200 bootstrap samples. Once this list has been compiled, both forward and backward manual selection procedures for model parameters will be conducted for the one-stage and two-stage analyses of the primary and secondary outcomes, provided sufficient data are available. The overlap between the parameters selected by the forward and backward selection procedures will highlight, from the list of potential treatment effect modifiers, parameters most likely to be true treatment effect modifiers.

This investigation involves a large number of planned analyses. However, this analysis of treatment modifiers of CBT is, to an extent, exploratory. As such, this investigation aims to identify potential relationships, and in so doing motivate future investigations specifically targeting the identified areas of interest. Consequently, and given that there is not a standard multiple testing approach currently recommended for IPD-MA, we will not adjust for multiple testing, although we reiterated that these analyses are, to an extent, exploratory. If methods are developed during the course of this project that are recommended as standard to account for multiple testing in IPD-MA, application of the methods will be examined.

The main analyses will be conducted as complete case analyses, that is, only those contributing data for all variables (outcome or explanatory) included in the model will be used in the MA. If the level of missing data for treatment effect modifiers or outcomes is large across the studies included in the meta-analysis, if possible, the effect of missing data on the conclusions of the analyses will be investigated by reconducting the analyses of the primary and secondary outcomes based on multiply imputed data sets, and the results compared with those obtained from the complete case analysis. This is in addition to the planned sensitivity analyses.

### Heterogeneity, bias and study quality

This investigation employs a random effects approach to analyses. As such, in two-stage analyses, statistical heterogeneity will be examined using the $\tau^2$ (which provides an estimate of between-study variance) and $I^2$ statistics (which provides the proportion of total variance that is due to 'true' heterogeneity in treatment effects, interpretation as stated in the Cochrane Handbook[25]). Additionally, the p value for the $\chi^2$ test for heterogeneity, along with visual inspection of forest plots, will be assessed. In one-stage analyses, heterogeneity can be assessed through the variance of the study-level random treatment effect and through examination of the coefficients for fixed study membership terms.

If substantial heterogeneity is observed between results from different groups of studies, data across heterogeneous groups of studies will not be pooled, and the demographics and characteristics of the differing groups of studies will be compared in an attempt to identify differences that could cause the heterogeneity. Potential evidence of substantial heterogeneity in two-stage analyses is defined in this investigation as some combination of (1) significant p value for $\chi^2$ test for heterogeneity in intervention tests at level of 0.10, (2) $I^2$ statistic greater than 50% (representing substantial heterogeneity), (3) visual inspection of the forest plot to identify heterogeneity (Cochrane Handbook, section 9.5.2[26]). Potential evidence of substantial heterogeneity in one-stage analyses will be given by comparison of models with and without the study-level random treatment effect and fixed study membership terms. If evidence exists of heterogeneity not accounted for through the proposed model structure, use of additional terms (fixed effects or study-level random effects) will be examined and noted.

Risk of bias in each study will be assessed using the Cochrane Collaboration risk of bias tool. Analyses and results will be interpreted in light of the risk of bias of the studies included in the meta-analysis.

Publication bias (and other selection bias/small study effects) will be investigated through inspection of contour-enhanced funnel plots and appropriate statistical tests for funnel plot asymmetry.[11 13 16] Assessment of publication bias will only be undertaken for analyses containing 10 or more trials (due to the low power of the assessments for analyses containing small numbers of trials).

Analyses of the data will clearly report the proportion of individuals within each study for which IPD could be obtained, as well as the numbers of studies which were deemed eligible to be included in the meta-analysis, but which did not supply any data. Interpretation of overall strength of the evidence regarding modifiers of patient response to treatment examined in this evidence synthesis will be appraised in light of relevant assessment of IPD integrity, availability and risk of bias.

### Subgroup and sensitivity analysis
#### Sensitivity analyses

For both primary and secondary outcomes, where conversion between scales has occurred, sensitivity analyses will be conducted that (1) remove studies with transformed data, analysing only data recorded on the main scale, and (2) use standardised scores across studies[18] (unless standardised scores have been used in the main analysis).

There were a range of ways in which some treatment effect modifiers could be coded in this investigation. As such, sensitivity analyses will also be conducted investigating whether the method of coding particular modifiers effected the results. Sensitivity analyses will also be conducted to investigate the unavailability of IPD (comparing results from two-stage MA that combines results from study-specific IPD analyses, with AD extracted from study reports, to the main analysis based only on IPD), and to investigate changes in treatment effect over time (primary and longitudinally measured secondary outcome one-stage and two-stage analyses will be reconducted including an interaction term between treatment and time).

#### Subgroup analyses

We will conduct subgroup analyses contrasting non-blind versus single-blind trials to examine the effect of masking of outcome assessments. Any analyses where outcomes have been transformed onto a common scale will also be subgrouped as data originally recorded on the common scale versus data transformed onto the common scale. In one-stage analyses, subgrouping will be achieved by interacting the coefficient of interest with the grouping variable. In two-stage analyses, results will be pooled from studies belonging to each subgroup.

### Patient and public involvement

Throughout this project, we have taken care to involve key stakeholders in the design of the research. The research team contains a PPI representative. Additionally, secondary outcomes and treatment effect modifiers were identified in part through consultation meetings with service users with psychosis and clinical psychologists and CBT therapists working with clients with psychosis in secondary care settings in the UK.

Results of this research will be disseminated to stakeholders via a range of methods including the preparation of an information pack for service users and carers, which will be made freely available online.

### ETHICS AND DISSEMINATION

This IPD-MA will be conducted in line with current recommendations of secondary analysis of IPD data (eg, ref [5]). Specific ethics approval for the IPD-MA is not required, as the objectives of the IPD-MA are consistent with those of the original trials, and do not violate the condition of consent under which the data were collected. Throughout, anonymised data will be sought from study authors. Information from data owners should not include any personally identifiable information. Data and accompanying documentation will be held on a secure server by the research team. Findings of this research will be submitted for publication in high-impact peer-reviewed journals, promoted to relevant stakeholders and presented to relevant research communities at international conferences.

### DISCUSSION

NICE and other clinical guidelines worldwide recommend CBT as an intervention for the treatment and management of psychosis and schizophrenia[1]; however, AD-MA to date has reported heterogeneous treatment effects between studies. This heterogeneity may be attributable to certain as yet unidentified patient or intervention-specific characteristics that influence the clinical effectiveness of CBT. This IPD-MA examines a range of potential treatment effect modifiers for CBT for patients with schizophrenia spectrum diagnoses, for the primary outcome overall psychotic symptoms severity, as well as a range of secondary outcomes commonly targeted by CBT for psychosis. The treatment effect modifiers and outcomes investigated in this project are wide ranging; however, given the lack of current research in this area, it is hoped that this exploratory IPD-MA will provide guidance for future focused research. This investigation is required to establish the effectiveness of CBT for psychosis across a range of different populations, and will provide evidence to improve healthcare for patients with schizophrenia spectrum diagnoses.

**Author affiliations**
[1]Department of Health Data Science, University of Liverpool, Liverpool, UK
[2]Psychosis Research Unit, Greater Manchester Mental Health NHS Foundation Trust, Manchester Academic Health Science Centre, Manchester, UK
[3]Complex Trauma and Resilience Research Unit, Greater Manchester Mental Health NHS Foundation Trust, Manchester Academic Health Science Centre, Manchester, UK
[4]Health Methodology Research, University of Manchester, Manchester, UK
[5]National Collaborating Centre for Mental Health, London, UK
[6]Department of Biostatistics and Health InformaticsInstitute of Psychiatry, Psychology and Neuroscience, King's College London, London, UK
[7]Division of Psychology and Mental Health, School of Health Sciences, University of Manchester, Manchester Academic Health Science Centre, Manchester, UK

**Acknowledgements** We thank all those involved in the PPI consultations that helped shape this research project.

**Contributors** FV, CTS and AM conceived the idea for the review, and are project managing the systematic review and meta-analysis. GD provided input in the initial stages of the development and design of the project. EL provided patient and public involvement of the project. TK will be involved in future dissemination of

research. CTS and MS drafted the statistical analysis plan for the project with help from RE, and (along with XL) input into data collection, cleaning and analysis. All authors contributed to the development of the idea and drafting and revision of the manuscript. All authors gave approval for the manuscript to be submitted.

**Funding** This project was funded by the National Institute for Health Research (NIHR) Health Technology Assessment Programme (project number 15/187/05).

**Disclaimer** The views expressed are those of the author(s) and not necessarily those of the NIHR or the Department of Health and Social Care.

**Competing interests** We recognise that one member of our team may be regarded as having a vested interest in CBT (AM is the only coapplicant actively involved in CBT training, leading of trial grants and receiving royalties from CBT texts or books).

**Patient and public involvement** Patients and/or the public were involved in the design, or conduct, or reporting, or dissemination plans of this research. Refer to the Methods section for further details.

**Patient consent for publication** Not required.

**Provenance and peer review** Not commissioned; externally peer reviewed.

**ORCID iDs**
Maria Sudell http://orcid.org/0000-0002-7919-4981
Catrin Tudur-Smith http://orcid.org/0000-0003-3051-1445
Richard Emsley http://orcid.org/0000-0002-1218-675X
Anthony Morrison http://orcid.org/0000-0002-4389-2091
Filippo Varese http://orcid.org/0000-0001-7244-598X

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
