## [Reviewer comments · BMJ Open]

ARTICLE DETAILS

TITLE (PROVISIONAL)	Protocol for Individual Participant Data Meta Analysis of randomised control trials of patients with psychosis, to investigate treatment effect modifiers for CBT versus treatment as usual or other psychosocial interventions
AUTHORS	Sudell, Maria; Tudur-Smith, Catrin; Liao, Xiaomeng; Longden, Eleanor; Dunn, Graham; Kendall, Tim; Emsley, Richard; Morrison, Anthony; Varese, Filippo

VERSION 1 – REVIEW

REVIEWER	Emma Motrico Universidad Loyola Andalucía
REVIEW RETURNED	30-Oct-2019

GENERAL COMMENTS	Thank you for inviting me to review the paper "For whom is Cognitive Behavioural Therapy (CBT) for psychosis most effective? Protocol for an IPD meta-analysis of randomised control trials comparing CBT versus standard care and other psychosocial interventions (Cognitive Behaviour Therapy for Psychosis: Individual Modifiers of Patient Response to Treatment)". Comments: 1) First, I suggest a clearer and shorter title.2) In the objectives of IPD-MA, you must provide an explicit statement of the questions being addressed with reference, as applicable, to participants, interventions, comparisons, outcomes, and study design (PICOS). Include any hypotheses that relate to particular types of participant-level subgroups. I suggest you move "treatment comparisons" and "treatment effect modifiers" to the method section.3) Obtaining IPD is time-consuming and contact with the researchers responsible for the original trials is usually required. Please, in the data collection and processes, describe the process "in the case of no response".
---

REVIEWER	Dr Anthony James University of Oxford UK
REVIEW RETURNED	08-Nov-2019

GENERAL COMMENTS	I have reviewed the paper and think it is well crafted and a sensible attempt to cover an important area. I do worry about two things - the number variables to be tested and why is it necessary to include open trials, with all the problems these entail. I am sure the open trials will not be included in the main analysis, but why include them at all? The authors have attempted to answer the question about the number of variables to be tested.
--

	The team is very well equipped to do this analysis and has expert statistical advice on board. I look forward to seeing the final result, which is an important area to tackle. I recommend for publication.
--	--

VERSION 1 – AUTHOR RESPONSE

Reviewer: 1

Reviewer Name: Emma Motrico

Institution and Country: Universidad Loyola Andalucía

Please state any competing interests or state 'None declared': None Declared

Please leave your comments for the authors below

Thank you for inviting me to review the paper "For whom is Cognitive Behavioural Therapy (CBT) for psychosis most effective? Protocol for an IPD meta-analysis of randomised control trials comparing CBT versus standard care and other psychosocial interventions (Cognitive Behaviour Therapy for Psychosis: Individual Modifiers of Patient Response to Treatment)".

Comments:

1) First, I suggest a clearer and shorter title.

As discussed above in the response to an editor comment, the title has been reworded and shortened

2) In the objectives of IPD-MA, you must provide an explicit statement of the questions being addressed with reference, as applicable, to participants, interventions, comparisons, outcomes, and study design (PICOS).

Statement of the primary and secondary objectives have been reworded to better highlight participants, interventions, comparisons, outcomes and study design, for example the primary outcome now reads "The primary objective of the IPD-MA of RCTs is to identify treatment effect modifiers for CBT or CBT+ vs Treatment As Usual (TAU) or other psychosocial interventions on overall psychotic symptoms severity as measured by PANSS scores [16, 17] in patients with schizophrenia-spectrum diagnoses."

Include any hypotheses that relate to particular types of participant-level subgroups.

As this analysis is exploratory, as the number of planned analyses is large, and as little information currently exists concerning the magnitude or direction of possible effects of these treatment modifiers (or participant level subgroups), we do not attempt to form hypotheses concerning the impact of treatment modifiers on the effect of treatment on any listed outcome. As such, we have not listed hypotheses that relate to particular types of participant level subgroups, as we hope that this research will provide initial evidence to allow future research to form hypotheses concerning participant level subgroups.

I suggest you move "treatment comparisons" and "treatment effect modifiers" to the method section.

Treatment comparisons and treatment effect modifiers sections have been moved to the methods section, see earlier response to editor comment.

3) Obtaining IPD is time-consuming and contact with the researchers responsible for the original trials is usually required. Please, in the data collection and processes, describe the process “in the case of no response”.

To address the case of no response, we have included the following paragraph in the “Data Collection and Processes” section:

“In the case of no response to data requests (defined as a minimum of 4 contact attempts with no response), details of the study would be stated as “non-acquired data”. Details concerning the number and proportion of studies and individuals for which IPD has been obtained will be stated in reports of analyses. Additionally, we have planned a sensitivity analysis to assess the impact of non-acquired data on results of the project, which will combine obtained IPD, with AD from studies not supplying IPD.”

Reviewer: 2

Reviewer Name: Dr Anthony James

Institution and Country: University of Oxford UK

Please state any competing interests or state ‘None declared’: None

Please leave your comments for the authors below

I have reviewed the paper and think it is well crafted and a sensible attempt to cover an important area.

I do worry about two things - the number variables to be tested and why is it necessary to include open trials, with all the problems these entail. I am sure the open trials will not be included in the main analysis, but why include them at all? The authors have attempted to answer the question about the number of variables to be tested.

The team is very well equipped to do this analysis and has expert statistical advice on board.

I look forward to seeing the final result, which is an important area to tackle.

I recommend for publication.

We acknowledge that the number of planned analyses in this project, and the number of variables (treatment effect modifiers) to be examined is high. However, we note that little research has been completed in this area. We intend this research to be, to an extent, exploratory, motivating further focussed research in the area. We have stated this in the data analysis section, but have now also included section reiterating this point in the discussion, which reads: “The treatment effect modifiers and outcomes investigated in this project are wide ranging, however given the lack of current research in this area, it is hoped that this exploratory IPD-MA will provide guidance for future focussed research.”. We also aim to reiterate this point in all future publications of analysis results.

We acknowledge the issues linked with analysis of open i.e. “unblinded” trials. However, in research involving patients with psychosis it is common that both the patient and the clinician has knowledge of the intervention allocation. In line with the Cochrane risk of bias tool, we have measured and recorded risk of bias for blinding of participants and personnel, and blinding of outcome assessment. We note that if we excluded trials where blinding of participants and personnel is classed as high risk of bias, the number of trials eligible for inclusion in the IPD-MA would be severely reduced. As such, we aim to highlight the issue of high risk of bias due to blinding of participants and personnel in publication of analyses, whilst noting this risk of bias is difficult to avoid in this field.

VERSION 2 – REVIEW

REVIEWER	Emma Motrico Universidad Loyola Andalucia, Spain.
REVIEW RETURNED	11-Feb-2020

GENERAL COMMENTS	All comments have been addressed satisfactorily.
--

REVIEWER	Dr Anthony James University of Oxford, UK
REVIEW RETURNED	10-Feb-2020

GENERAL COMMENTS	The revision has improved the paper and I believe the queries raised have been addressed satisfactorily and, in particular, the protocol conforms to the standard conventions for systemic reviews and meta-analyses.
---